# Measurement of Uncertainty in Prediction of No-Reflow Phenomenon after Primary Percutaneous Coronary Intervention Using Systemic Immune Inflammation Index: The Gray Zone Approach

**DOI:** 10.3390/diagnostics13040709

**Published:** 2023-02-13

**Authors:** Ebru Ozturk, Kerim Esenboga, Alparslan Kurtul, Mustafa Kilickap, Ergun Karaagaoglu, Jale Karakaya

**Affiliations:** 1Department of Biostatistics, Faculty of Medicine, Hacettepe University, Ankara 06230, Turkey; 2Department of Cardiology, Faculty of Medicine, Ankara University, Ankara 06590, Turkey; 3Department of Cardiology, Faculty of Medicine, Hatay Mustafa Kemal University, Antakya 31060, Turkey

**Keywords:** statistical detection, uncertainty, systemic immune inflammation index, no-reflow phenomenon, primary percutaneous coronary intervention

## Abstract

Systemic immune-inflammation index (SII), which is a good predictive marker for coronary artery disease, can be calculated by using platelet, neutrophil, and lymphocyte counts. The no-reflow occurrence can also be predicted using the SII. The aim of this study is to reveal the uncertainty of SII for diagnosing ST-elevation myocardial infarction (STEMI) patients who were admitted for primary percutaneous coronary intervention (PCI) for the no-reflow phenomenon. A total of 510 consecutive acute (STEMI) patients with primary PCI were reviewed and included retrospectively. For diagnostic tests which are not a gold standard, there is always an overlap between the results of patients with and without a certain disease. In the literature, for quantitative diagnostic tests where the diagnosis is not certain, two approaches have been proposed, named “grey zone” and “uncertain interval”. The uncertain area of the SII, which is given the general term “gray zone” in this article, was constructed and its results were compared with the “grey zone” and “uncertain interval” approaches. The lower and upper limits of the gray zone were found to be 611.504–1790.827 and 1186.576–1565.088 for the grey zone and uncertain interval approaches, respectively. A higher number of patients inside the gray zone and higher performance outside the gray zone were found for the grey zone approach. One should be aware of the differences between the two approaches when making a decision. The patients who were in this gray zone should be observed carefully for detection of the no-reflow phenomenon.

## 1. Introduction

Due to the clinical success of percutaneous coronary intervention (PCI), which was initially developed by Grüntzig et al. [1] for ST-elevation myocardial infarction (STEMI), primary PCI is the gold standard. The advantage of PCI has been found in that, according to thrombolytic research, it is more successful in repairing thrombolysis in myocardial infarction (TIMI) three-grade flow, which leads to lower mortality [2]. In the setting of STEMI, in spite of the immediate opening of the culprit artery in the absence of spasm, dissection, or angiographic epicardial vessel obstruction, the resumed blood flow may be insufficient. This situation is called “no-reflow ” [3]. Factors leading to no-reflow and the incriminated situations include prolonged myocardial ischemia, consecutive endothelial dysfunction in the coronary microcirculation, and failure of proper blood flow restoration at this level [2]. The no-reflow situation occurs in more than 20% of patients undergoing primary PCI for STEMI and in less than 2% of elective procedures; moreover, it is a predictor of short- and long-term mortality [4].

The use of different diagnostic modalities, including contrast echocardiography, myocardial magnetic resonance imaging, and radionuclide imaging techniques (such as single-photon emission computed tomography—SPECT—and positron emission tomography—PET) can be of use to precisely evaluate STEMI patients who develop no-reflow [5]. However, diagnosing the no-reflow phenomenon with these methods is quite time-consuming. Moreover, early detection of no-reflow is crucial since the results of the primary PCI will change by considering treatment of no-reflow and preventative measures [6]. Therefore, we sought to evaluate a direct assessment of no-reflow based on angiographic visual evaluation.

The systemic immune-inflammation index (SII) is the platelet count multiplied by the neutrophil–lymphocyte ratio (NLR), which is proposed to evaluate the status of inflammation and immunothrombosis of patients. A higher SII indicates poorer outcomes in cancer patients [7]. In recent studies, it has been reported that the SII may play a key role in the prognosis of cardiovascular diseases such as chronic heart failure [8] and no-reflow after PCI in patients with STEMI [9].

Decision-making in medicine mostly ends up in a binary way, such that the subject either has a disease or not, or should either take treatment or not. The clinician should rule in or rule out some diseases, and to achieve this goal, diagnostic tests are widely used. For quantitative diagnostic tests with a binary outcome, an optimal cut-off point is specified to discriminate which subjects have the disease and which do not. Moreover, this cut-off point may not discriminate subjects perfectly, which indicates false positive or negative counts. The decision about the disease is not certain at intermediate values in a given range. To deal with this uncertainty, a middle inconclusive area, which is indicated as the gray zone in this article, is proposed [10] and adapted for categorical data [11], ordinal data [12], and quantitative data [13]. Test results falling into this inconclusive area lead to a poor diagnosis as to whether the condition under consideration is present or absent. In some cases, it is important to reveal the subjects who are in the gray zone for decision-making. For instance, advanced methods may be required to diagnose subjects in the gray zone.

In this study, we aim to construct and specify the bounds of the gray zone for the no-reflow phenomenon for patients with STEMI treated by primary PCI using SII. The main aim of this approach is to classify subjects into the categories normal flow, no-reflow and gray zone. In this gray zone, subjects have been classified as having neither normal flow nor no-reflow. This might mean early detection of no-reflow; in this case, the necessary precautions might be taken. To construct a gray zone, two approaches, known as “grey zone” [13] and “uncertain interval” [14], were used. (In order to prevent confusion about the “grey zone” approach and the “gray zone”, we used the “gray zone” for the general term of this uncertain area. On the other hand, the “grey zone” approach is one of the algorithms applied in this article.) Thus, we will compare the results and performance of those two approaches.

## 2. Materials and Methods

### 2.1. Dataset

The dataset used in this study was collected retrospectively to reveal the effect of SII on no-reflow for primary PCI patients as in the first study [9]. In this study, we used the same dataset to construct the gray zone. All acute STEMI patients who received primary PCI within 12 h of symptom onset between October 2015 and January 2020 were initially included in this retrospective cross-sectional study. STEMI was defined as symptoms of myocardial ischemia accompanied by a persistent elevation of the ST segment on an electrocardiogram (the presence of a new left bundle branch block, or the presence of ≥1 mm ST-segment elevation in the inferior lead or ≥2 mm ST-segment elevation in the anterior chest lead occurring in at least two continuous leads) and the subsequent release of biomarkers of myocardial necrosis [15]. Exclusion criteria for patient recruitment were as follows: (1) the presence of any chronic inflammatory or autoimmune disease, including rheumatologic disorders, (2) hematological disorders, (3) known malignancy, (4) end-stage liver and renal failure, (5) cardiogenic shock, (6) pain-to-balloon durations longer than 12 h, and (7) treatment with fibrinolytic drugs. After exclusion, a total of 510 patients was included in the final analysis. Approval was given by the Human Research Ethics Committee of the Ankara University Faculty of Medicine according to the Declaration of Helsinki and good clinical practice on 28 September 2020 (Approval code: i8-535-20). Due to the design of this study being retrospective, informed consent was disregarded.

### 2.2. Coronary Procedures

At the time of STEMI diagnosis, 300 mg of aspirin was given to all individuals, along with a single loading dose of 600 mg of clopidogrel, 180 mg of ticagrelor, or 60 mg of prasugrel. Standard procedures were used to perform the baseline coronary angiography. As soon as the decision was made to perform a cardiac intervention, patients received a 50–70 unit/kg intravenous bolus dose of unfractionated heparin. Further, 6 F or 7 F guiding catheters were used for all primary PCI operations. An Artis Zee Floor workstation (Siemens Medical Solution, Erlingen, Germany) was used to analyze the cineangiography. The categories of the angiographic thrombus burden are represented in Table 1 [16].

The visual classification of the TIMI flow grade on post-procedure angiograms served as the basis for the definition of the no-reflow phenomenon because the acquired angiographic films were not long enough to calculate myocardial blush grading. Two experienced interventional cardiologists evaluated TIMI flow grades without knowing the clinical information of the patients. The following definitions apply to the TIMI flow grades: Grade 0 denotes the complete absence of flow following the culprit lesion. In Grade 1, the occlusion site is reached by the contrast material, but the entire artery is not completely opacified. Normal coronary flow is evaluated as Grade 3, while completely opacified arteries distal to the obstruction point and slower-than-normal flow are evaluated as Grade 2 [17]. The criteria for a successful PCI were TIMI flow Grade 3 for the treated infarct-related artery (IRA) and residual stenosis of less than 20%. Upon the final angiography, patients with anterograde TIMI flow ≤ 2 who did not exhibit dissection, thrombus, spasm, or distal embolization were found to have no-reflow [18]. Based on their final angiographic TIMI flow rates, the patients were classified into two groups: ta normal reflow and a no-reflow group. The occurrence of more than one lesion with a stenosis of over 50% in more than one major epicardial coronary artery or one of its major branches far from the IRA was referred to as multi-vessel disease.

### 2.3. Laboratory Measurements

Before the coronary procedure, all patients had venous blood samples drawn from their antecubital veins after admission. A detailed list and features of biochemical tests are given in Table 3. The total peripheral platelet count multiplied by the neutrophil–lymphocyte ratio (NLR) was used to compute the SII [6]. The platelet–lymphocyte ratio (PLR) and neutrophil–lymphocyte ratio (NLR) were calculated as platelet count and neutrophil count divided by lymphocyte count. The estimated glomerular filtration rate (GFR) was also calculated [19].

Clinical recommendations were followed for preoperative critical medicines [15]. By taking into account the conditions that patients had, the cardiologists of the coronary intensive care unit chose the types and dosages of angiotensin-converting enzyme inhibitors or angiotensin II receptor antagonists, beta-blockers, and statins. The operator scheduled the usage of glycoprotein IIb/IIIa inhibitors based on the patient’s clinical state. Other peri-procedural methods were used at the operator’s discretion. All echocardiographic measures were performed using a GE ViVidE7 ultrasound machine (GE Healthcare, Piscataway, NJ, USA) with a 3.5 MHz transducer within 24 h after the surgeries. Following the advice of the American Society of Echocardiography, the Simpson method was used to calculate the left ventricular ejection fraction (LVEF).

### 2.4. Statistical Analysis

Descriptive statistics are presented as median (first and third quartiles) for continuous variables and frequency and relative frequency (%) for categorical variables. While categorical variables were compared using Fisher’s exact test, continuous variables were compared using the Mann–Whitney U test. For presenting the discriminative abilities of the SII, NLR and PLR, the receiver operating characteristic (ROC) curve was plotted. The area under the ROC curve was evaluated.

#### Determination of the Gray Zone

To construct the boundaries of the gray zone, two approaches described in the literature, the “grey zone” [13] and “uncertain interval” [14] approaches were applied.
(i)*Grey Zone Approach*Coste and Pouchot [13] define the boundaries of the gray zone area via negative and positive likelihood ratios. To find the likelihood ratios, the following steps are used:
(a)Specify the pre-test probabilities. The pre-test probabilities are chosen based on the prevalence of no-reflow in the sample. The number of patients in the no-reflow and normal flow groups was 110 and 400, respectively. Therefore, the pre-probabilities of having no-reflow and normal flow were taken as 110/510 = 0.216 and 400/510 = 0.784, respectively.(b)Specify the post-test probabilities. The post-test probabilities are specified as 0.6 and 0.95 for positive predictive and negative predictive values, respectively.(c)The positive likelihood ratio (LR+) and the negative likelihood ratio (LR-) are calculated using these test probabilities and were found to be 5.455 and 0.191 by using Equation (Equation 1).
(1)$$\begin{array}{cc} \; & \mathrm{The}\;\mathrm{post-test}\;\mathrm{odds}\;\mathrm{of}\;\mathrm{having}\;a\;\mathrm{disease}=\mathrm{The}\;\mathrm{pre-test}\;\mathrm{odds}\;\mathrm{of}\;\mathrm{having}\;a\;\mathrm{disease}\times LR(+)\\ \; & \;\;\;\;\;\;\;\;\;\;\;\;\;\;\;\;\;\;\mathrm{and}\\ \; & \mathrm{The}\;\mathrm{post-test}\;\mathrm{odds}\;\mathrm{of}\;\mathrm{not}\;\mathrm{having}\;a\;\mathrm{disease}=\mathrm{The}\;\mathrm{pre-test}\;\mathrm{odds}\;\mathrm{of}\;\mathrm{not}\;\mathrm{having}\;a\;\mathrm{disease}\times \frac{1}{LR(-)} \end{array}$$(ii)*Uncertain Interval Approach*An alternative approach to finding the inconclusive area has been proposed [14]. This approach depends on a different trichotomization method by using the two decision thresholds based on pre-selected values of sensitivity and specificity in this uncertain area. In this study, the pre-selected values are chosen as 0.55 for both sensitivity and specificity in the gray zone by considering the default values.

To compare these two algorithms, the number of subjects in the gray zone and the width of the gray zone are reported. Further, the accuracy, sensitivity, and specificity inside and outside the gray zone were calculated by using the “caret” package [20].

The descriptive statistics and two group comparisons were analyzed with IBM SPSS 23 [21]. Moreover, the graphics and gray zone boundaries were obtained with the “ggplot2” package [22] and “UncertainInterval” package [23] in R [24].

## 3. Results

The baseline characteristics and comparisons of the normal flow and no-reflow groups are represented in Table 2. In total, 400 (78.431%) and 110 (21.569%) patients were in normal flow and no-reflow, respectively. There were no statistically significant differences with regard to sex and traditional cardiovascular risk factors among the normal flow and no-reflow groups.

The laboratory measurements and comparisons of the normal flow and no-reflow groups are represented in Table 3. Patients who had no-reflow had significantly higher platelet and neutrophil counts, but significantly lower lymphocyte counts. For the indexes of NLR, SII and PLR, patients who had no-reflow had significantly higher values compared to patients with normal flow.

In order to observe the index performances of SII, NLR, and PLR, the ROC curves and areas under the ROC curves were evaluated; see Figure 1. The area under the ROC curve was found to be 0.839, 0.744, and 0.69 for SII, NLR and PLR, respectively. Hence, the performance of the SII is higher than the NLR and PLR when the area under the ROC curve was considered. As a result, the boundaries of the gray zone were constructed based on SII to obtain better performance.

The results for the identification of approaches to identifying the gray zone are presented in Figure 2, Figure 3 and Figure 4 and in Table 4. The lower and upper limits of the gray zone for the uncertain interval approach were 1186.576 and 1565.088. The number of patients in between these limits was 40, and 9 of them had no-reflow. Moreover, the patients who had lower values than the lower limit of the gray zone were classified as having normal flow, whereas the patients who had higher values than the upper limit of the gray zone were classified as having no-reflow. Accuracy, sensitivity, and specificity were calculated based on this classification. Moreover, the 95% confidence intervals for these performance measures are also reported. The accuracy, sensitivity and specificity values were 0.804 (0.766–0.838), 0.732 (0.691–0.771) and 0.824 (0.787–0.856), respectively, outside the gray zone, while for the grey zone approach, the boundaries of the gray zone were found as 611.504 and 1790.827. The number of patients who fall in this area is 220, and 36 of them were diagnosed with no-reflow. Accuracy, sensitivity, and specificity were found to be 0.824 (0.776–0.864), 0.878 (0.836–0.911) and 0.806 (0.756–0.847), respectively, in the outer area of the gray zone.

It can clearly be seen that the grey zone approach had wider limits of the gray zone boundaries compared to the uncertain interval approach. Moreover, as is expected due to the size of the gray zone, the number of patients who are in the gray zone is lower in the uncertain interval approach compared to the grey zone approach. The accuracy and sensitivity of the grey zone approach were higher than the uncertain interval approach, while its specificity was lower compared to the uncertain interval approach. There was no statistical difference in accuracy (0.804 versus 0.824, *p* = 0.127) and specificity (0.824 versus 0.806, *p* = 0.188) between the uncertain interval and grey zone approaches, respectively. The statistical difference was significant for sensitivity (0.732 versus 0.878, *p* < 0.001).

## 4. Discussion

The association between SII and the no-reflow phenomenon in acute STEMI patients who received primary PCI is a new topic to research and discuss [4,25]. It has been emphasized that baseline SII levels are independently related to acute STEMI patients who developed no-reflow after primary PCI and had considerably higher baseline SII levels. The no-reflow phenomenon is one of the most common primary PCI complications that might result in negative outcomes, including increased mortality [4,25]. It is still unclear exactly how the no-reflow phenomenon’s pathophysiological mechanisms work. The no-reflow phenomenon is secondary to vasoconstriction, platelet and leukocyte activation, enhanced oxidant production, formation of oxygen free radicals, and malfunction of the vascular endothelium, according to the generally acknowledged consensus in this context [26].

Restricting the outcomes of a decision to only binary classes might cause some problems; for this reason, considering the middle inconclusive area is recommended in the literature. There are different approaches to dealing with the middle inconclusive area. Feinstein [10] points out how the two zones specified by binary models are not sufficient for medical decision-making. The author also states that constructing three zones for decisions as either "yes", "inconclusive" or "no" is not unusual for both clinical and statistical approaches. Moreover, this is illustrated using the Bayesian approach and likelihood ratios for creatinine kinase for diagnosing myocardial infarction.

No-reflow, also known as TIMI flow grade < 3 and myocardial blush grade < 3, presents as abnormal epicardial blood flow even if coronary occlusion has been relieved. An electrocardiogram may show sustained ST-segment elevation in addition to hemodynamic instability, anginal symptoms, and other clinical signs. No-reflow has a multifactorial mechanism. The primary causes might be a microvascular obstruction resulting in distal embolization of thrombus or debris, microvascular spasm, intravascular plugging from platelet microthrombi or leukocytes, or ischemic reperfusion injury [4,27]. Risk factors for no-reflow include being female, being aged >65, traditional cardiovascular risk factors (such as hypertension, diabetes, and hyperlipidemia), delayed presentation to the PCI center, baseline kidney insufficiency, and increased systemic inflammatory marker levels [28].

## 5. Conclusions

In this study, we applied the grey zone approach and uncertain interval approach by using SII to find out the boundaries of the gray zone. Therefore, we classified the patients into three groups. The patients who had higher SII values than the upper bound of the gray zone are in the no-reflow group, while those who had lower SII values than the lower bound of the gray zone are in the normal-reflow group. The patients in between the lower and upper limits of the gray zone were classified into neither the no-reflow group nor the normal flow group. Hence, we compared the performance of the grey zone and the uncertain interval approaches. In the grey zone approach, more subjects were inside the gray zone compared to the uncertain interval approach. Furthermore, the accuracy and sensitivity outside the gray zone were higher in the grey zone approach compared to the uncertain interval approach, contrary to specificity. In order to choose which approach is beneficial, both the performance measure outside the gray zone and the number of patients inside the gray zone were considered. As a final remark, the main contribution of this study can be summarized such that constructing a gray zone might lead clinicians to take precautions, for instance, primary stenting, avoidance of high-pressure post-dilatation, using an embolic protection device and pretreatment with intracoronary vasodilators for performing a saphenous vein graft PCI for the patients in the gray zone to prevent no-reflow.

## 6. Limitations

There are certain limitations to this study. The study design involved collecting retrospective data from a single center. The greatest drawback of the current investigation was the absence of specific diagnostic methods, such as myocardial contrast echocardiography or coronary magnetic resonance imaging in addition to angiographic evaluation to diagnose no-reflow. Another drawback is that the complete blood count test was obtained during primary PCI. Therefore, the operator applied intervention without knowing the SII. In addition, it may be advantageous to connect no-reflow with left ventricular function (pre-discharge) or with post-discharge clinical outcome in order to increase the reliability of the no-reflow definition (heart failure and mortality).

As no-reflow is a sign of a high thrombotic state, which is also influenced by plaque burden and its characteristics as well as patient risk factors and location and length of the lesion, and also regarding the timing of the presentation, it would be beneficial to evaluate plaque burden and its characteristics using intracoronary imaging methods, namely intravascular ultrasound and optical coherence tomography.

In both the grey zone and uncertain interval approaches, we found different ranges that directly affect the performances of those outside the grey zone. The wider the range of the gray zone, the higher the number of patients inside the gray zone. It might be beneficial to hold one of the performance measures in the gray zone constant to find the narrowest range, which might indicate the best approach. It is crucial to understand that constructing a gray zone may provide clinicians with the means to construct a decision-making algorithm. For instance, they might apply and find more information from other biomarkers or more sophisticated tests to rule in or rule out the disease for the patients in the gray zone. Moreover, for the decision-making algorithm, the different gray zones might be constructed in different subgroups. In the context of this study, factors such as age (greater than 60) or blood glucose level (greater than 12 mmol/L) might be related to no-reflow [6]. In this sample, there is no statistical significance regarding those factors, so the boundaries of the gray zone are constructed from all observations in the sample. Therefore, for further studies, those drawbacks should be considered.

## Figures and Tables

**Figure 1 diagnostics-13-00709-f001:**
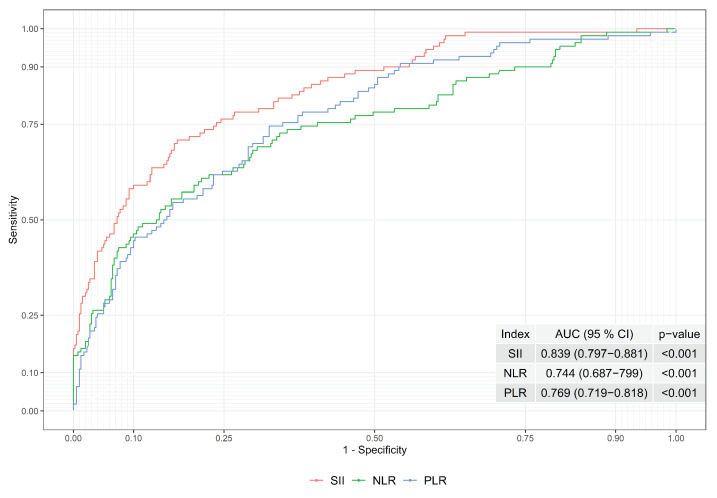
ROC Curves of Indexes of SII, NLR and PLR.

**Figure 2 diagnostics-13-00709-f002:**
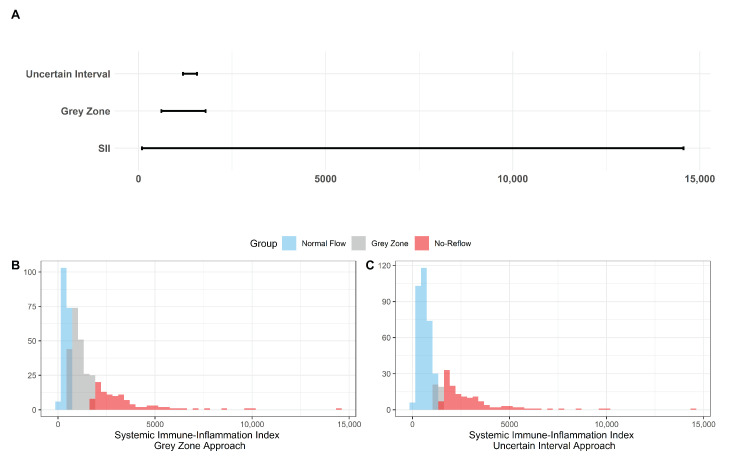
The Boundaries of Gray Zone. (**A**) The range of SII (minimum and maximum values were recorded as 92.2 and 14562.2) is given in the bottom line while the boundaries of the gray zone are represented in the middle and the top lines with respect to the grey zone and the uncertain interval approaches. (**B**) Histogram of SII with the representation of the 3 classes based on the grey zone approach as the normal flow, gray zone, and no-reflow. (**C**) Histogram of SII with the representation of the 3 classes based on the uncertain interval approach as the normal flow, gray zone, and no-reflow.

**Figure 3 diagnostics-13-00709-f003:**
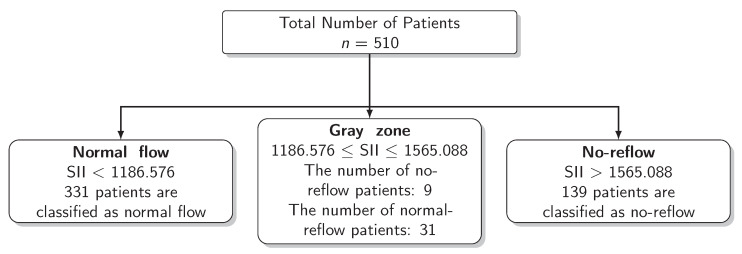
Scheme of the Uncertain Interval Approach.

**Figure 4 diagnostics-13-00709-f004:**
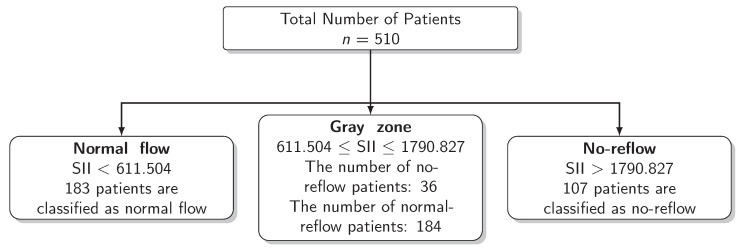
Scheme of the Grey Zone Approach.

**Table 1 diagnostics-13-00709-t001:** Grades of Angiographic Thrombus Burden.

Grades of Angiographic Thrombus Burden	
Grade 0	No thrombus
Grade 1	Thrombus may exist
Grade 2	The thrombus’ largest size is less than half a vessel in diameter
Grade 3	The largest dimension is between one half and two vessels in diameter
Grade 4	There are more than two vessel diameters in the largest dimension
Grade 5	The thrombus completely occludes the vessel

**Table 2 diagnostics-13-00709-t002:** Baseline characteristics of the patients.

Variable	Normal Flow (*n* = 400)	No-Reflow (*n* = 110)	Test Statistic	*p*-Value
Age	62 (54–70)	62.50 (53–70)	0.039	0.969
Male	314 (78.5)	77 (70)	NA	0.074
History of hypertension	237 (59.25)	57 (51.818)	NA	0.191
History of diabetes	151 (37.75)	34 (30.909)	NA	0.218
Smoking	133 (33.250)	29 (26.364)	NA	0.203
History of hyperlipidemia	124 (31)	31 (28.182)	NA	0.640
Family history of coronary heart disease	85 (21.25)	29 (26.364)	NA	0.301
Prior stroke	7 (1.75)	5 (4.545)	NA	0.145
Time from pain to intervention (>6 h)	89 (22.25)	19 (17.273)	NA	0.293
Prior aspirin therapy	134 (33.5)	21 (19.091)	NA	0.003
Prior statin therapy	109 (27.25)	20 (18.182)	NA	0.063
Prior clopidogrel therapy	32 (8)	7 (6.364)	NA	0.687
In hospital statin therapy	400 (100)	110 (100)	NA	NA
Gp IIb/IIIa inhibitor therapy	41 (10.25)	15 (13.636)	NA	0.306
Killip class (3–4)	33 (8.25)	19 (17.273)	NA	0.012
Anterior infarction location	188 (47)	62 (56.364)	NA	0.086
Left ventricular ejection fraction	45 (40–45)	45 (40–50)	−0.931	0.352

**Table 3 diagnostics-13-00709-t003:** The laboratory measurements of the patients.

Variable	Normal Flow (*n* = 400)	No-Reflow (*n* = 110)	Test Statistic	*p*-Value
Platelet count (×10${}^{9}$/L)	239 (204–278)	354.5 (284–377)	−10.464	<0.001
Neutrophil count (×10${}^{9}$/L)	6.2 (4.915–8.295)	11.63 (7.18–15.12)	−8.843	<0.001
Lymphocyte count (×10${}^{9}$/L)	2.005 (1.5–2.955)	1.695 (1.13–2.34)	−3.541	<0.001
Neutrophil to lymphocyte ratio	2.903 (1.767–5.074)	7.13 (3.456–11.874)	−7.826	<0.001
Systemic immune-inflammation index	690.791 (413.911–1161.518)	2066.281 (1190.72–3493.188)	−10.893	<0.001
Platelet to lymphocyte ratio	113.231 (80.776–161.786)	190.788 (138.153–299.18)	−8.63	<0.001
Serum glucose (mg/dL)	118 (99–160.5)	129 (105–171)	−1.819	0.069
Serum creatinine (mg/dL)	0.88 (0.75–1.06)	0.92 (0.77–1.09)	−1.233	0.218
C-reactive protein (mg/dL)	6.3 (2.6–18.3)	13.95 (5.9–63.6)	−4.885	<0.001
Peak cardiac troponin (ng/dL)	294.5 (40.6–3142.3)	773.9 (47.63–6702)	−1.681	0.093
Total cholesterol (mg/dL)	185 (160–221.5)	193.5 (152–236)	−0.347	0.728
Low-density lipoprotein cholesterol (mg/dL)	120 (94–146.5)	123 (89–163)	−0.619	0.536
High-density lipoprotein cholesterol (mg/dL)	39 (33–45)	39 (33–49)	−0.672	0.501
Triglyceride (mg/dL)	134 (90–199)	122.5 (88–160)	−1.744	0.081
Hemoglobin (g/dL)	14.5 (12.9–15.6)	13.7 (11.5–15.1)	−3.237	0.001
Glomerular filtration rate (mL/min/1.73 m${}^{2}$)	88 (72.5–94)	84.5 (65–92)	−1.546	0.122
Type of intervention				
*Direct Stenting*	36 (9)	11 (10)	NA	0.713
*Balloon angioplasty before stenting*	364 (91)	99 (90)		
Thrombus aspiration	7 (1.75)	2 (1.818)	NA	>0.999
Intra-aortic balloon pump use	6 (1.5)	0 (0)	NA	0.346
Total stent length (mm)	30 (22–46)	33 (24–56)	−1.238	0.216
Stent diameter (mm)	2.75 (2.5–3)	2.875 (2.5–3)	−0.495	0.621
Multi-vessel disease	155 (38.75)	39 (35.455)	0.398	0.528

**Table 4 diagnostics-13-00709-t004:** The Results for Approaches to Gray Zone Identification.

Approaches	Gray Zone Limits		Patients Inside Gray Zone			Patients Outside Gray Zone		
	Lower Limit– Upper Limit	Size	Total	No-Reflow	Normal Flow	Accuracy	Sensitivity	Specificity
Uncertain Interval	1186.576–1565.088	378.5125	40	9	31	0.804	0.732	0.824
Grey Zone	611.504–1790.827	1179.323	220	36	184	0.824	0.878	0.806

## Data Availability

The data are not publicly available due to privacy or ethical restrictions.

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
