# Peer review of "Measurement of Uncertainty in Prediction of No-Reflow Phenomenon after Primary Percutaneous Coronary Intervention Using Systemic Immune Inflammation Index: The Gray Zone Approach"

_diagnostics, 2023, doi:10.3390/diagnostics13040709_

Round 1
Reviewer 1 Report
Although the study has potential in being interesting, it must be better restructured and developed. In this regard, I suggest the following:
1. Keywords must reflect the main characteristic words of the paper (usually reflected also by the title) in the best way to increase the paper's relevance and chances to be find when searching it after key words. So, for the actual title, I suggest the following relevant keywords: statistical detection; uncertainty; systemic immune inflammation index; no-reflow phenomenon; primary percutaneous coronary intervention.
2. In the Abstract section please revise the usage of abbreviations since STEMI appears directly abbreviated. Moreover, in the entire manuscript please revise abbreviations as the Instructions for authors must be checked and respected “Acronyms/Abbreviations/Initialisms should be defined the first time they appear in each of three sections: the abstract; the main text; under the first figure or table. When defined for the first time, the acronym/abbreviation/initialism should be added in parentheses after the written-out form”. No Abbreviations section is mentioned in the Instructions for authors, so please proceed consequently and remove it.
3. In the Introduction section, which must be more developed, I recommend a complete revision. In the first two paragraphs, although the citations are correctly assigned, the word replacements in the original text are debatable and completely change the sentences` meaning. The no-reflow phenomenon should have been explained in a more detailed manner especially regarding the diagnostic methods. I suggest checking and referring to: https://doi.org/10.3390/diagnostics12040932 and https://doi.org/10.3390/jcm9092956
4. Also, please revise the last paragraph of the section Introduction and rephrase the aim of the paper, highlighting the novelty of the research/special aspects it brings to the field. The plan of the manuscript (L53-55) must be removed as it can be seen through the manuscript, and it is not part of the aim of the study.
5. L57-58 must be removed. No relevance. It is obvious what it must be described in the 2nd section.
6. L73-74. No/date of the approval (of Ethics Committee of Ankara University Faculty of Medicine) must be provided.
7. Subsection 2.1. Inclusion and exclusion criteria for patients’ recruitment must be provided.
8. In the data paragraph of material and methods section, please use references from ESC guidelines (https://doi.org/10.1093/eurheartj/ehx393 ) in order to give the STEMI definition/frame (for a better understanding).
9. L85-89. I suggest tabulating that text. It would be easier to follow.
10. Tables must be inserted as closest is possible to their first mention in the main text.
11. Please be more specific about the variables in the Table 2, specify whether they are mean values or not and provide their measuring units (where applicable). Also, please write in both tables the total number of the patients with and without no-reflow, respectively, as it gives a better understanding of the data.
12. L159-174 are providing information in duplicate with the tables. Reshape/remove, please. An information must be given once in a paper.
13. The tables can be enlarged on the entire width of the page, as MDPI draft allows it. Especially for Table 2, please proceed.
14. Set Table 3 according to MDPI Instructions for authors.
15. In the Discussion section please revise lines 228 to 231 in order to clarify the relation of citations with the present study.
16. Conclusion section is missing, as separate sectio. However, the text for it, is there. I suggest to add the title of 5. Conclusions above L273. A separate section of Conclusions is more relevant.
17. Again, no not forget removing Abbreviations section. CMRI Nardiac magnetic resonance imaging must be revised.
Author Response
We are grateful for your careful reading, suggestions and contributions.
- Keywords must reflect the main characteristic words of the paper (usually reflected also by the title) in the best way to increase the paper's relevance and chances to be find when searching it after key words. So, for the actual title, I suggest the following relevant keywords: statistical detection; uncertainty; systemic immune inflammation index; no-reflow phenomenon; primary percutaneous coronary intervention.
Response: We revised the keywords and added your suggestions in Lines 18-19.
- In the Abstract section please revise the usage of abbreviations since STEMI appears directly abbreviated. Moreover, in the entire manuscript please revise abbreviations as the Instructions for authors must be checked and respected “Acronyms/Abbreviations/Initialisms should be defined the first time they appear in each of three sections: the abstract; the main text; under the first figure or table. When defined for the first time, the acronym/abbreviation/initialism should be added in parentheses after the written-out form”. No Abbreviations section is mentioned in the Instructions for authors, so please proceed consequently and remove it.
Response: We reviewed your suggestions. We corrected the usage of abbreviations of STEMI in the Abstract Section Line 4.
- In the Introduction section, which must be more developed, I recommend a complete revision. In the first two paragraphs, although the citations are correctly assigned, the word replacements in the original text are debatable and completely change the sentences` meaning. The no-reflow phenomenon should have been explained in a more detailed manner especially regarding the diagnostic methods. I suggest checking and referring to: https://doi.org/10.3390/diagnostics12040932 and https://doi.org/10.3390/jcm9092956
Response: We changed the Introduction section in Lines 22-39.
- Also, please revise the last paragraph of the section Introduction and rephrase the aim of the paper, highlighting the novelty of the research/special aspects it brings to the field. The plan of the manuscript (L53-55) must be removed as it can be seen through the manuscript, and it is not part of the aim of the study.
Response: We rephrased the aim of the paper more clearly in Lines 62 -71. Thus, following your suggestions, we removed the manuscript plan.
- L57-58 must be removed. No relevance. It is obvious what it must be described in the 2nd section.
Response: We removed the Lines 57-58.
- L73-74. No/date of the approval (of Ethics Committee of Ankara University Faculty of Medicine) must be provided.
Response: The approval date and code of ethics committee were added in Line 90.
- Subsection 2.1. Inclusion and exclusion criteria for patients’ recruitment must be provided.
Response: Inclusion and exclusion criteria for patients’ recruitment were provided in Lines 76 - 89.
- In the data paragraph of material and methods section, please use references from ESC guidelines (https://doi.org/10.1093/eurheartj/ehx393 ) in order to give the STEMI definition/frame (for a better understanding).
Response: We clearly defined the STEMI in Lines 78 - 82.
- L85-89. I suggest tabulating that text. It would be easier to follow.
Response: We present the text in Lines 85-89 in Table 1.
- Tables must be inserted as closest is possible to their first mention in the main text.
Response: We inserted Tables as closest to their first mention in the main text.
- Please be more specific about the variables in the Table 2, specify whether they are mean values or not and provide their measuring units (where applicable). Also, please write in both tables the total number of the patients with and without no-reflow, respectively, as it gives a better understanding of the data.
Response: The units of variables are given for Table 3 (it is now Table 3 since we added a new Table). The number of patients with normal flow and no-reflow is given in Tables 2 and 3.
- L159-174 are providing information in duplicate with the tables. Reshape/remove, please. An information must be given once in a paper.
Response: We paraphrased and reshaped Lines 159-164 in Lines 179- 187.
- The tables can be enlarged on the entire width of the page, as MDPI draft allows it. Especially for Table 2, please proceed.
Response: The width of Tables is arranged.
- Set Table 3 according to MDPI Instructions for authors.
Response: The width of Tables is arranged.
- In the Discussion section please revise lines 228 to 231 in order to clarify the relation of citations with the present study.
Response: We removed the Lines 223 -231 to prevent this confusion.
- Conclusion section is missing, as separate section. However, the text for it, is there. I suggest to add the title of 5. Conclusions above L273. A separate section of Conclusions is more relevant.
Response: We divided the Conclusion section.
- Again, no not forget removing Abbreviations section. CMRI Nardiac magnetic resonance imaging must be revised.
Response: The Abbreviations section is removed.
Reviewer 2 Report
Authors present two non-Bayesian (non binary) methods for predicting no-reflow following PPCI, based on the Systemic immune inflammatory index derived from the blood count. Namely, they are: the gray zone approach and the Uncertain Interval Approach. They concluded that the Gray zone approach has better sensitivity (and diagnostic accuracy) at the price of a marginal reduction in specificity. Also, they found that the application of the systemic immune-inflammation index is a useful prediction tool in this setting.
However, I found a lot of objections – however, they appeared to be manageable.
The article suffers from inconsistent terminology and wording, where phrases such as uncertain area, "gray zone", gray zone and uncertain interval are sometimes used interchangeably. The best example is in the Abstract (lines 6-8 and around). That could produce confusion among readers and impair its' acceptance and perception.
The article's title, per se, is diluted and is not a good representation of the work done. Such a title is hard to understand at first glance, which would impair its' acceptance and apprehension to the readers. The title should be more concise and focused; I am just proposing one that I think would be more suitable: "Measurement of uncertainty in Prediction of No-Reflow Phenomenon After Primary Percutaneous Coronary Intervention using Systemic Immune Inflammation Index: The Gray zone approach".
Secondly, the article is exceptionally lengthy and needs significant shortening. Some data from results are repeated in the discussion and so forth.
Also, the article needs a clear conclusion (in a separate session). The conclusion is buried in the extensive discussion so that after reading it, one is not certain what is the final message of the article.
Study restrictions should take a separate section (last two paragraphs).
In preparing the revision, authors should correct wordings and sentences where their approach could be miss-perception as detection of no-reflow instead of prediction. And, in general: extensive shortening! Besides many similar examples, a sentence in lines 26-27 is redundant and irrelevant to the subject and should be deleted.
Finally, a touch of English medical lector, knowledgeable about the subject, would be welcome.
Author Response
We are grateful for your careful reading, suggestions and contributions
Authors present two non-Bayesian (non binary) methods for predicting no-reflow following PPCI, based on the Systemic immune inflammatory index derived from the blood count. Namely, they are: the gray zone approach and the Uncertain Interval Approach. They concluded that the Gray zone approach has better sensitivity (and diagnostic accuracy) at the price of a marginal reduction in specificity. Also, they found that the application of the systemic immune-inflammation index is a useful prediction tool in this setting.
However, I found a lot of objections – however, they appeared to be manageable.
- The article suffers from inconsistent terminology and wording, where phrases such as uncertain area, "gray zone", gray zone and uncertain interval are sometimes used interchangeably. The best example is in the Abstract (lines 6-8 and around). That could produce confusion among readers and impair its' acceptance and perception.
Response: We used The “gray zone” term in the article as a general term this uncertain area to prevent the confusion of “grey zone” algorithm. It is stated in Lines 8 - 11, 68 - 70.
- The article's title, per se, is diluted and is not a good representation of the work done. Such a title is hard to understand at first glance, which would impair its' acceptance and apprehension to the readers. The title should be more concise and focused; I am just proposing one that I think would be more suitable: "Measurement of Uncertainty in Prediction of No-Reflow Phenomenon After Primary Percutaneous Coronary Intervention using Systemic Immune Inflammation Index: The Gray Zone Approach".
Response: We changed the article’s title considering your suggestions.
- Secondly, the article is exceptionally lengthy and needs significant shortening. Some data from results are repeated in the discussion and so forth.
Response: We removed repeated parts in Results and Conclusion sections.
- Also, the article needs a clear conclusion (in a separate session). The conclusion is buried in the extensive discussion so that after reading it, one is not certain what is the final message of the article.
Response: We separated the conclusion part.
- Study restrictions should take a separate section.
Response: We divided study restrictions in separate section.
- In preparing the revision, authors should correct wordings and sentences where their approach could be miss-perception as detection of no-reflow instead of prediction. And, in general: extensive shortening! Besides many similar examples, a sentence in lines 26-27 is redundant and irrelevant to the subject and should be deleted.
Response: We revised the article. The article has gone English language editing. We corrected wordings and sentences, removed irrelevant and redundant sentences. We shortened the Results and Discussion sections. We removed the Lines 26 – 27.
- Finally, a touch of English medical lector, knowledgeable about the subject, would be welcome.
Response: The article has gone English language editing.
Reviewer 3 Report
It is a significant area of interest to readers however neutrophils and other cells are various other factors that can drive their value and has multiple biases to conclude it has a correlation with no-reflow. As no-reflow is a sign of a high thrombotic state which also is influenced by plaque burden and its characteristics as well as patient risk factors and location and length of the lesion, also the timing of the presentation. The author also needs to define the aim clearly and objectively
Author Response
We are grateful for your careful reading, suggestions and contributions
It is a significant area of interest to readers however neutrophils and other cells are various other factors that can drive their value and has multiple biases to conclude it has a correlation with no-reflow. As no-reflow is a sign of a high thrombotic state which also is influenced by plaque burden and its characteristics as well as patient risk factors and location and length of the lesion, also the timing of the presentation. The author also needs to define the aim clearly and objectively.
Response: In the Limitations sections of the article, we mentioned other factors which are related with no-reflow in Lines 287 -292. For plaque burden issue, you are very right, thank you very much for your attention. As you mentioned, it would be great if we could evaluate plaque burden and its characteristics with intracoronary imaging methods, namely intravascular ultrasound and optical coherence tomography. We have added this to the Limitations section in Lines 274 – 278. We also define the aim of study more clearly in the Introduction section in Lines 62 -71.
Round 2
Reviewer 1 Report
The authors responded to my requests.
Author Response
We are grateful for your careful reading, suggestions and contributions
The article has gone to English language editing for major revision by MDPI. the English Editing Certificate is in the attachment. Moreover, we also revised and corrected some spellings for this minor revision. The blue-colored texts are the changes for the major revision while the red-colored texts are the changes for the minor revision. Moreover, we also added references in the Introduction Section Lines 28 and 68.
Reviewer 3 Report
Thank you the necessary changes
Author Response
We are grateful for your careful reading, suggestions and contributions
The article has gone to English language editing for major revision by MDPI. the English Editing Certificate is in the attachment. Moreover, we also revised and corrected some spellings for this minor revision. The blue-colored texts are the changes for the major revision while the red-colored texts are the changes for the minor revision.